# The Influence of Photoactive Heterostructures on the Photocatalytic Removal of Dyes and Pharmaceutical Active Compounds: A Mini-Review

**DOI:** 10.3390/nano10091766

**Published:** 2020-09-07

**Authors:** Alexandru Enesca, Luminita Andronic

**Affiliations:** Product Design, Mechatronics and Environmental Department, Transilvania University of Brasov, Eroilor 29 Street, 35000 Brasov, Romania; andronic-luminita@unitbv.ro

**Keywords:** semiconductors, heterostructures, photocatalysis, dyes, pharmaceutical compounds

## Abstract

The diversification of pollutants type and concentration in wastewater has underlined the importance of finding new alternatives to traditional treatment methods. Advanced oxidation processes (AOPs), among others, are considered as promising candidate to efficiently remove organic pollutants such as dyes or pharmaceutical active compounds (PhACs). The present minireview resumes several recent achievements on the implementation and optimization of photoactive heterostructures used as photocatalysts for dyes and PhACs removal. The paper is focused on various methods of enhancing the heterostructure photocatalytic properties by optimizing parameters such as synthesis methods, composition, crystallinity, morphology, pollutant concentration and light irradiation.

## 1. Introduction

The increase of the world population, especially in the last 50 years has stimulated the industrial growth and water consumption. However, with the rapid development of industrialization, the presence of organic pollutants in water and environment is now an important problem, creating hazardous threats to the ecological system [1,2,3].

Dyes and pharmaceutical compounds represent two important classes of organic pollutants affecting the water and human life quality [4,5]. The contamination with dyes substances such as rhodamine B (RhB), malachite green (MG), methyl orange (MO), etc. have raised serious issues related to human health. Most of the dyes are considered harmful due to the high toxicity and carcinogenicity induced by their nonbiodegradable aromatic structure [6,7,8]. Pharmaceutically active compounds (PhACs), such as tetracycline (TC), ciprofloxacin (CIP), triclosan (TCS), carbamazepine (CBZ), salicylic acid (SA), ibuprofen (IBS) and sulfamethazine (SMZ), etc., are helpful to cure human diseases but the pollution from metabolized or partially metabolized pharmaceutical wastes is already recognized as a hazard [9,10,11].

Advanced oxidation processes (AOPs) represent an energy-efficient and green approach suitable for environmental remediation, particularly optimized for wastewater treatment due to the use of solar energy as a driving force to initiate the oxidation reactions [12,13,14]. The photocatalysis is considered as an advanced oxidation process, using stand-alone or coupled semiconductors as well as other composite materials [15,16,17].

Breakthrough heterostructures that benefit from the full range of solar light have attracted the attention of many scientists, the UV, visible or near infrared photoexcitation depending on their band gaps energy and ability to generate (super)oxidative species able to destroy organic pollutants [18,19]. Owing to the controllable band structures and efficient electron-hole separation, the heterostructured photocatalyst exhibits a superior performance to their individual components [20,21,22].

The synergistic effect of coupling photocatalysis with other techniques in order to improve the pollutant degradation efficiency was studied by many research groups. Coupling adsorption with photocatalysis has shown promising results on RhB [23] and MG [24] dyes removal. Encouraging results were obtained on PhACs [25,26] and dyes [27] removal by coupling biodegradation with photocatalysis. Enhanced mineralization conversion efficiencies were obtained by coupling flocculation with photocatalysis [28], ultrasound with photocatalysis [29] and plasma with photocatalysis [30].

The present minireview resumes several recent achievements on the implementation and optimization of photoactive heterostructures used as photocatalysts for dyes and PhACs removal. Many other papers which are not included here have the potential to contain highly innovative work. The paper is focused on the influence of crystallinity, morphology, pollutant concentration, irradiation time and light spectra on the photocatalytic activity of various heterostructures. This minireview make a comparative evaluation of the radiation intensity and spectra required for organic pollutant removal based on the heterostructure charge transport mechanism. The photocatalytic efficiency refers to partial or total pollutant degradation according to the information’s provided in the literature. The lack of standardization regarding the photocatalysis experimental parameters has, as a consequence, the presence of various scientific papers containing data difficult to compare.

## 2. Heterostructure Mechanisms for Photocatalytic Application

Photocatalysis is a chemical process mediated by one or more semiconductors, which under irradiation increase the reduction and oxidation (redox) reactions rate based on charge carriers’ generation [31,32]. When the chemical potential of the electrons from the conduction-band (CB) is between +0.5 and −2.0 V versus the normal hydrogen electrode (NHE), they act as reductants due to the strong oxidizability [33,34,35]. The photocatalytic process is characterized by three main steps: (i) electron-hole pair’s generation due to the light absorption, (ii) the diffusion of photoexcited charge carriers on the semiconductor surface and (iii) the redox reaction on the semiconductor surface [36,37,38].

As presented in Figure 1, during the photocatalytic degradation of organic pollutants, the photogenerated electrons and holes are trapped by dissolved O_2_ and H_2_O [39,40]. Highly reactive oxygen species, including superoxide (O_2_^–^) and hydroxyl radicals (OH), are developed under light irradiation [41,42]. If the organic pollutants contain only carbon, oxygen and hydrogen atoms, it can be degraded to by-products or even completely mineralized (CO_2_ and H_2_O). However, the major part of dyes and PhACs contains also other atoms, and the mineralization occurs with the formation of additional products (based on Cl^−^, NH_4_^+^, SO_3_^2−^, etc.) [43,44].

Photoactive heterostructures are considered as composite materials developed by using at least two components/compounds with similar or different chemical nature and able to develop charge carriers during the light irradiation [45,46,47]. The literature outlines four typical heterostructures mechanisms (p-n junctions, type II heterostructures, Schottky junctions and Z-scheme heterostructures) mostly used in photocatalytic studies.

The p-n junctions (see Figure 2) mimics the photovoltaic cell mechanism but with the difference that the charge carriers are used in redox reactions. Between the p-type and n-type semiconductors, an interfacial region is formed and the electrons from the CB of the p-type semiconductor migrate to the n-type semiconductor. The holes follow the reverse direction, from the n-type semiconductor VB to the p-type semiconductor VB forming a space-charge region. Due to the charge carrier’s diffusion, the space-charge region will have an established built-in potential directing the electrons and holes flow in opposite directions [48,49,50].

In the type-II (see Figure 3), the potential difference between the semiconductors will easily separate electrons and holes, preventing subsequent carrier recombination. The reason consists on the fact that the CB and VB of one semiconductor are higher than that of the other semiconductor, which assures an effective separation with holes and electrons at different sides of the heterojunction [51,52,53].

The Schottky junctions (see Figure 4) represented by the semiconductor–metal interface, can induce an effective way to reduce charge carriers’ recombination and extend the spectral light absorption of the semiconductor. One of the most important features is represented by the localized surface plasmonic resonance (LSPR) effect, which assures the visible light absorption and the excitation of active electrons/holes pairs. The LSPR takes place when the work function of the metal surpasses that of the semiconductor, developing a positive space-charge region on the semiconductor surface as well as an upwards of the semiconductor band [54,55,56].

The Z-scheme (see Figure 5) consists of two connected photocatalytic systems, with the advantage of endowing charge carriers with stronger reduction/oxidation properties.

These systems can use shuttle redox mediators, solid-state electron mediators or no electron mediators. The photogenerated electrons from the second photocatalytic systems will migrate to the electron medium, in order to recombine with the photogenerated holes from the first photocatalytic system. In this way, the opposite electrons and holes are involved in the reduction and oxidation reactions, which assure a highly efficient charge separation [57,58,59].

A relatively new type of heterostructure mechanism is represented by S-scheme (see Figure 6), where the photogenerated electrons from photocatalyst I with lower conduction band (CB) potential will combine the photogenerated holes from photocatalyst II with higher valence band (VB) potential based on the driving force of the internal electric field, thereby inducing the spatial separation of charge carriers under irradiation, considering the conditions of ensuring the strong redox ability [60].

## 3. Photocatalytic Organic Pollutants Removal by Heterostructures

### 3.1. Dyes

Dyes are substances with complex chemical structures involving aromatic cycles and functional groups [61,62,63]. There are several classifications such as acid dyes, alkaline dyes, azo dyes, sulphur dye, etc., most of them considering the composition and the solubility in water. Between others, the literature mentions the following representative dyes: RhB, methylene blue (MB), MO and MG. These dyes molecules are characterized by aromatic structures with high chemical stability toward traditional wastewater treatment processes. Table 1 includes representative studies regarding photocatalysts materials (composition, synthesis method, morphology and crystallinity) and photocatalytic parameters (pollutant concentration, radiation type, intensity and time), influencing the dyes photocatalytic removal process (efficiency and rate constant).

The RhB photocatalytic removal using g-C_3_N_4_-based heterostructures obtained by ultrasonication [64,65] and hydrothermal [66,67] methods were investigated in the presence of Vis irradiation. The Bi_12_TiO_20_/g-C_3_N_4_ [64] and Bi_3__.84_W_0.16_O_6.24_ (BWO)/g-C_3_N_4_ [65] heterostructures were evaluated in similar photocatalytic conditions (10 mg/L RhB and 50 min irradiation period), but with different light intensities (500 W for Bi_12_TiO_20_/g-C_3_N_4_ and 100 W for Bi_3__.84_W_0.16_O_6.24_ (BWO)/g-C_3_N_4_). Using smaller radiation intensity, the Bi_3__.84_W_0.16_O_6.24_ (BWO)/g-C_3_N_4_ heterostructure exhibits higher photocatalytic efficiency (99.8%) with lower energy consumption than that of Bi_12_TiO_20_/g-C_3_N_4_ (97%). Both heterostructures follow the Z-scheme mechanism and have similar crystalline structures, but different morphologies (sheet-like for Bi_3__.84_W_0.16_O_6.24_ (BWO)/g-C_3_N_4_ and multilayer for Bi_12_TiO_20_/g-C_3_N_4_). A similar observation is valid for TiO_2_/g-C_3_N_4_ [66] and g-C_3_N_4_/ZnO [67] obtained by the hydrothermal method. In this case, the g-C_3_N_4_/ZnO heterostructure achieves 98.5% photocatalytic efficiency toward RhB after 70 min of 300 W Vis light irradiation. The TiO_2_/g-C_3_N_4_ reaches only 85% after 100 min of Vis light irradiation (500 W). The g-C_3_N_4_/ZnO benefit from the large rods shape interfaces, which improve the energy conversion due to the larger active surface. In TiO_2_/g-C_3_N_4_, there was tensile stress induced by TiO_2_ and compressive stress from C_3_N_4_, with detrimental influence on the heterostructure interfacial area. However, using the same irradiation scenario (Vis, 500 W for 100 min) but employing WS_2_/BiOBr heterostructure [68], the photocatalytic efficiency of RhB (20 mg/L) has increased up to 95%, due to high Vis absorbance attributed to both partners. Even if the photocatalytic efficiency is lower compared with g-C_3_N_4_/ZnO, the RhB concentration was double.

Solvothermal method was used to develop NiO/BiOI S-scheme heterostructure [60], Bi_2_MoO_6_/Bi_5_O_7_Br/TiO_2_ Z-scheme heterostructure [69] and Bi_2_MoO_6_/Fe_3_O [70] type II heterostructure. The optimum energy consumption corresponds to NiO/BiOI reaching 90% RhB (5 mg/L) photocatalytic efficiency in 60 min of irradiation with 300 W Vis light source. The NiO/BiOI is characterized by porous morphology and follows the S-scheme mechanism, in which the electrons from CB of BiOI migrate towards NiO in the junction interface due to the formation of an internal electric field. The Bi_2_MoO_6_/Fe_3_O heterostructure has higher photocatalytic efficiency (99.5%) but after a longer irradiation period (120 min) with 350 W Vis light source. However, in this case, the RhB concentration was 20 mg/L, which means that even if the energy consumption was higher, the quantity of pollutant removed from the aqueous solution was also bigger. The Bi_2_MoO_6_/Fe_3_O is a type II heterostructure with flower-like morphology, who benefit from the surface oxygen vacancies, which serve as electron-trapping sites, and have an important role in the charge transmission through the heterostructures by facilitating the migration of the bulk electron-hole. Higher energy consumption corresponds to Bi_2_MoO_6_/Bi_5_O_7_Br/TiO_2_, which follows a Z-scheme mechanism and exhibits 73.43% photocatalytic efficiency of RhB (10 mg/L) after 180 min of irradiation with 500 W Vis light source.

Bi_2_MoO_6_/Fe_3_O type II heterostructure [70] was employed to investigate MB removal in the same photocatalytic conditions as RhB removal. However, in the case of MB, the photocatalytic efficiency has increased up to 93.81%, indicating that this heterostructure can be optimized for a particular type of pollutant molecule. Ta_3_B_2_@Ta_2_O_5_ [71] heterostructure was obtained by in situ growth method, and the photocatalytic properties were tested in the same conditions as Bi_2_MoO_6_/Fe_3_O (180 min irradiation period, 500 W Vis light source). The photocatalytic efficiency was lower (80%) compared with Bi_2_MoO_6_/Fe_3_O, but the quantity of RhB removal was higher, considering that the initial dye concentration was 5 times bigger (50 mg/L). Concluding, for the same amount of energy consumption the quantity of pollutant removal is significantly higher. The Ta_3_B_2_@Ta_2_O_5_ heterostructure is characterized by irregular shape particles and contains high crystalline Ta_3_B_2_ and Ta_2_O_5_. The heterostructure mechanism corresponds to Schottky junction and has the advantage of the metallic transition zone, which improves the visible light absorption and the electron collector facilitating the charge transfer from CB of Ta_2_O_5_ to surface.

The g-C_3_N_4_/ZnO heterostructure [67] previously presented for RhB removal was also tested for MB (10 mg/L) removal in the same photocatalytic conditions (70 min, 300 W Vis light source). In this particular study, there was no significant difference regarding the photocatalytic efficiency values for both dyes (98% MB and 98.5% RhB) meaning that the heterostructure can efficiently remove different organic dye molecules. A similar experiment was done with BiPO_4−*x*_/B_2_S_3_ [72] using half of the MB concentration (5 mg/L) and 300 W Vis light intensity. The photocatalytic efficiency has reached 98% after 360 min of irradiation, which represents 5 times more energy consumption for a lower pollutant concentration. The BiPO_4−*x*_/B_2_S_3_ has a sheets morphology containing monoclinic BiPO_4_ and orthorhombic Bi_2_S_3_ and follows a Z-scheme pathway. The lower photocatalytic efficiency can be a consequence of ultraviolet light heterostructure exposure before the photocatalytic experiment, inducing electrons migration during the conversion of oxygen atoms in oxygen vacancy who may act as the recombination centre of photoinduced charge carriers.

UV light was used to evaluate the photocatalytic performance of spherical particles MnFe_2_O_4_/rGO [73] and flower-like Ag/hybridized 1T-2H MoS_2_/TiO_2_ [74] after 60 min of irradiation. Both heterostructures exhibit similar efficiencies (~97%) but at different MB concentrations. The MnFe_2_O_4_/rGO heterostructure obtained by coprecipitation method requires only 40 W light intensity to remove 97% of MB (10 mg/L) due to the reduced graphene oxide insertion (rGO), which induces a decrease of MnFe_2_O_4_ crystallite size from 21 to 18 nm and increase the junction interfacial area. The Ag/hybridized 1T-2H MoS_2_/TiO_2_ obtained by chemical reduction exhibits a dual Schottky junction and Z-scheme mechanisms, and was able to remove 96.8% of MB (20 mg/L) using 235 W UV light source. Even if the MB concentration was double, the energy consumption has increased almost 6 times which make this system less energy efficient. Another heterostructure obtained by coprecipitation method and tested under UV irradiation using 30 mg/L MB aqueous solution was ZnAl_2_O_4_/Bi_2_MoO_6_ [75]. The heterostructure exhibits a sheet-like morphology containing koechlinite Bi_2_MoO_6_ and gahnite ZnAl_2_O_4_, and follow a type II mechanism in which the VB potential of Bi_2_MoO_6_ is higher than that of ZnAl_2_O_4_ favouring the formation of superoxidative species. After 180 min of UV irradiation (100 W), the photocatalytic efficiency was 86.36%, due to the hinder effect of the ZnAl_2_O_4_ on the Bi_2_MoO_6_ light absorption through the suspension solution.

A comparative study regarding the influence of light spectra on the photocatalytic efficiency at low MB concentration (1 mg/L) was done using CuO–TiO_2_ heterostructure [76], obtained by ultrasonication. The TiO_2_ fibres sizes were in the range of 150–500 nm with CuO particles on the surface. The study indicates that CuO–TiO_2_ follow a p-n heterostructure mechanism with superior photocatalytic properties during UVc irradiation when the rate constant was 0.135 min^−1^, almost ten times higher than that in the presence of Vis irradiation. Consequently, after 45 min of UVc (96 W) irradiation, 99% of MB was removed and the same value was obtained by Vis (240 W) irradiation but after 240 min.

The influence of 300 W Vis light source on the MO photocatalytic removal was tested using LaNiO_3_/TiO_2_ S-scheme [77], ZnFe_2_O_4_/SnS_2_ p-n [78] and WO_3_/g-C_3_N_4_ Z-scheme [79] heterostructures. ZnFe_2_O_4_/SnS_2_ heterostructure obtained by the solvothermal method has the lowest energy consumption being able to remove 99% of MO (50 mg/L) in just 20 min. The ZnFe_2_O_4_/SnS_2_ heterostructure exhibits irregular particles shape, and the charges transport mechanism is based on the photogenerated electrons migration into the CB of SnS_2_ from the CB of ZnFe_2_O_4_. The electron-hole recombination is avoided, due to the photogenerated holes transit from VB of SnS_2_ to VB of ZnFe_2_O_4_. LaNiO_3_/TiO_2_ and WO_3_/g-C_3_N_4_ show similar photocatalytic activity toward MO (10 mg/L), reaching 100% photocatalytic efficiency in 150 min (LaNiO_3_/TiO_2_) and 93% in 120 min (WO_3_/g-C_3_N_4_). The LaNiO_3_/TiO_2_ heterostructure with irregular particles morphology was obtained by sol–gel technique and contains anatase/rutile TiO_2_ as well as crystalline LaNiO_3_, both being able to develop separate electron-hole pairs under irradiation. The charge carrier’s concentration increases and induces a certain potential difference able to enhance the photocatalytic activity of the S-scheme. The photocatalytic efficiency decreases at 92% when MO concentration is double (20 mg/L), due to the heterostructure limitation to form enough oxidative and superoxidative species in a short period. The porous morphology and small g-C_3_N_4_ sheet provide larger surface area and homogenous spread active sites, housing the photochemical reactions and facilitating the mass transport through the WO_3_/g-C_3_N_4_ heterostructure.

The photocatalytic activity decreases when 90 W UVc light was used to irradiate Ag_2_Mo_1−x_W_x_O_4_ (Ag_2_WO_4_/Ag_2_MoO_4_) [80] rod-shaped heterostructure, obtained by the microwave-assisted hydrothermal method. Even if the MO concentration was relatively low (5 mg/L), the photocatalytic efficiency after 140 min was 45% due to the available light spectra. The photocatalytic activity can be increased by using an extended light spectrum, considering that both Ag_2_WO_4_ and Ag_2_MoO_4_ have good Vis absorbance.

Highly concentrated MG (50 mg/L) aqueous solution was used to evaluate the photocatalytic properties of TiO_2_/WO_3_ [81] and CdS@ZnS@ZnO [82] heterostructures. Under Vis irradiation, the TiO_2_/WO_3_ hollow sphere heterostructure shows a better energy performance reaching 98% photocatalytic efficiency after 60 min using a 300 W light source. The TiO_2_/WO_3_ composite can efficiently use the visible light to the electronic structure of composite and the quantum effect arising from small particle size. CdS@ZnS@ZnO spherical shape heterostructure obtained by the hydrothermal method has bigger energy consumption and requires 400 W Vis light source for 180 min to remove 32.5 mg/L (65%) from the 50 mg/L MG initial concentration. However, the situation is drastically changed under UV (125 W) light where the photocatalytic efficiency increases at 95% only after 30 min of irradiation. The CdS@ZnS@ZnO follows a type II mechanism, where the VB and CB potential of ZnO are higher than ZnS allowing the photogenerated holes to be transferred on VB of ZnS. When the CdS was added, the photogenerated electrons could migrate from the CB of ZnS to the CB of CdS and further transferred on the CB of ZnO. It seems reasonable to consider that UV scenario is more energy efficient, but the sustainability issue must be underlined (the available UV sunlight spectra on the Earth surface is rather limited).

La_2_CuO_4_-decorated ZnO [83] and MgFe_2_O_4_/Bi_2_MoO_6_ [84] photocatalytic properties based on Z-scheme mechanism were tested after 120 min of irradiation. The results show that MgFe_2_O_4_/Bi_2_MoO_6_ with plate morphology reaches 97% photocatalytic efficiency for MG removal (20 mg/L initial concentration) using a 300 W Vis light source. The La_2_CuO_4_-decorated ZnO heterostructure obtained by in situ extraction exhibits lower photocatalytic efficiency (91%), but at a higher MG concentration (25 mg/L) and in the presence of 125 W Vis light source. Based on the rate constant provided by the authors, the La_2_CuO_4_-decorated ZnO have a higher rate constant corresponding to MG removal due to the La_2_CuO_4_ ability to convert visible light and to reduce oxygen molecules to superoxide radicals.

To sum up, the most energy-efficient photocatalytic systems correspond to type II mechanism heterostructures with energy consumption between 62 (47.5 mg/L MG removal by CdS@ZnS@ZnO) and 300 Wh (49 mg/L MG removal by TiO_2_/WO_3_). Higher energy consumption is attributed to Z-scheme heterostructure (Bi_2_MoO_6_/Bi_5_O_7_Br/TiO_2_), which uses 1500 Wh to remove 7.3 mg/L of RhB and 7.64 mg/L of MO. However, the correlation between the heterostructure mechanism and the energy consumption must consider other factors as well (heterostructure composition, pollutant type and concentration, radiation type, etc.).

### 3.2. Pharmaceutical Active Compounds

The increase of PhACs concentration raises essential issues to traditional wastewater treatment [85,86,87] and requires the involvement of novel processes such as photocatalysis [88,89,90]. The literature mentions a high number of PhACs investigated in AOP experiments, between the most representative are: TC, CIP, TCS, CBZ, SA, IBS and SMZ. Table 2 includes representative studies regarding photocatalysts materials (composition, synthesis method, morphology and crystallinity) and photocatalytic parameters (pollutant concentration, radiation type, intensity and time) influencing the PhACs photocatalytic removal process (efficiency and rate constant).

The photocatalytic removal of TC under Vis light as irradiation source was evaluated using g-C_3_N_4_-based heterostructures. Three heterostructures (BN/B-doped-g-C_3_N_4_ [91], WO_3_/g-C_3_N_4_ [79] and g-C_3_N_4_-decorated ZrO_2−*x*_ [92]) following the Z-scheme mechanism were irradiated with a 300 W Vis light source in order to study the photocatalytic activity toward 10 mg/L TC solution. g-C_3_N_4_-decorated ZrO_2−*x*_ with tube morphology obtained by anodic oxidation and physical vapour deposition exhibits lower energy consumption, requiring only 60 min to reach 90.6% photocatalytic efficiency. This result was influenced by the abundant defects states and lattice disorder, allowing ZnO_2−*x*_ to extend the absorption range to visible spectral region. The interfacial band bending and directed build-in electric field present in the band structure induce an increase of the charge carriers’ mobility and concentration. The sheets-like BN/B-doped-g-C_3_N_4_ heterostructure obtained by in situ growth exhibits 88.1% photocatalytic efficiency after 60 min of irradiation. The small difference in the photocatalytic efficiency can be induced by the influence of boron doping concentration on the charge carrier mobility. The BN will canalize the photogenerated charges without recombination, which helps electrons to move to the active sites on photocatalyst surface. The boron nitride has a hexagonal structure and does not significantly affect the crystal structure of B-doped-g-C_3_N_4_. The highest photocatalytic efficiency (97%) corresponds to WO_3_/g-C_3_N_4_ heterostructure, who was presented during MO-dedicated subsection. However, the energy consumption is significantly higher, considering that the irradiation time is three times longer (180 min) just to gain an extra 6.4% at the overall photocatalytic efficiency.

Using a combined type II and Z-scheme mechanisms, the photocatalytic activity of Ag_3_PO_4_/Co_3_ (PO_4_)_2_/g-C_3_N_4_ [93] heterostructure was tested using a 10 mg/L TC solution. The Ag_3_PO_4_/Co_3_ (PO_4_)_2_/g-C_3_N_4_ heterostructure was obtained by precipitation method and exhibited flower-like morphology. The photocatalytic efficiency (88%) is similar to BN/B-doped-g-C_3_N_4_ (88.1%) but at longer irradiation period (120 min, 300 W Vis light source), which requires higher energy consumption. The dual electron transfer induced by the combined mechanisms is still a subject of optimization, in order to have a rational design of ternary heterostructure that facilitates multilevel electron transfer.

The TC solution with 20 mg/L concentration was used to study the photocatalytic activity of two Z-mechanism heterostructures (MoS_2_/g-C_3_N_4_/Bi_24_O_31_Cl_10_ [94] and CuBi_2_O_4_/Bi_2_WO_6_ [95]), one type II heterostructure (La(OH)_3_/BiOCl [96]) and one p-n heterostructure (WS_2_/BiOBr [68]). La (OH)_3_/BiOCl type II heterostructure with sheets morphology obtained by microwave method shows very low energy consumption and reach 85% photocatalytic efficiency in 60 min of irradiation with a 5 W Vis source. The photocatalytic activity is attributed to the shorted mitigation distance (due to the small BiOCl thickness ~18 nm) followed by the photogenerated electrons during the migration process. The two heterostructures based on Z-mechanism show photocatalytic efficiencies above 90% in the presence of 300 W Vis light source. MoS_2_/g-C_3_N_4_/Bi_24_O_31_Cl_10_ has lower energy consumption and higher photocatalytic efficiency, reaching 97.5% in 60 min while CuBi_2_O_4_/Bi_2_WO_6_ requires double irradiation period (120 min) to remove 93% of TC. The higher MoS_2_/g-C_3_N_4_/Bi_24_O_31_Cl_10_ photocatalytic efficiency is explained by the ability to work as a dual Z-scheme ternary heterostructure, in which each component will generate charge carriers under irradiation. Consequently, the charge carriers’ concentration will be higher in the ternary heterostructure compared with the binary heterostructure. The p-n heterostructure (WS_2_/BiOBr) previously described in correlation with RhB dye was tested for two pharmaceutical molecules, and the photocatalytic efficiency was 96% for TC and 92% for CIP (20 mg/L). In order to reach these values, the WS_2_/BiOBr requires 100 min of irradiation using high-intensity (500 W) Vis light source. At the same CIP concentration and irradiation period, the BiOCl/CQDs/rGO [97] sheets-like heterostructure exhibits 87% photocatalytic efficiency using a lower intensity (300 W) Vis light source. Light-harvesting enhancement by CQDs (carbon quantum dots) and rGO (reduced graphene oxide) was not a crucial factor for increasing the photocatalytic activity, but they have a significant influence on accelerating the charge transfer and suppressing the recombination of photogenerated charge carriers.

An analogue evaluation was done using two heterostructures (ZnIn_2_S_4_/BiPO_4_ [98] and AgI/Bi_2_MoO_6_/AgBi (MoO_4_)_2_ [99]) obtained by hydrothermal method and tested using low TC (5 mg/L) and high TC (40 mg/L) concentrations. Both heterostructures were irradiated with Vis light for 90 min. AgI/Bi_2_MoO_6_/AgBi (MoO_4_)_2_ with sheets morphology was tested in 5 mg/L TC solution using a 400 W Vis source, and the photocatalytic efficiency was 91.9%. Contrary, the ZnIn_2_S_4_/BiPO_4_ with flower-like morphology was tested at a higher TC concentration (40 mg/L) but using lower Vis light source intensity (300 W), and the photocatalytic efficiency was 84%. These results show that by optimizing the heterostructure composition, it is possible to remove higher (eight times) pollutant concentrations with lower energy consumption. AgI/Bi_2_MoO_6_/AgBi (MoO_4_)_2_ follows a Z-scheme mechanism and has the advantage of ternary structure with multiple charge carriers’ injection. ZnIn_2_S_4_/BiPO_4_ is a type II heterostructure who benefit from the high specific surface (~100 m^2^/g) due to the dandelion-like microflower structure.

Z-scheme (PVPbiochar@ZnF_2_O_4_/BiOBr [100]) and S-scheme (LaNiO_3_/TiO_2_ [77]) mechanisms were used to evaluate the CIP photocatalytic removal under Vis light (300 W). The PVPbiochar@ZnF_2_O_4_/BiOBr heterostructure with sheets-like morphology was obtained by the solvothermal method and contains tetragonal BiOBr and spinel ZnFe_2_O_4_. The Z-scheme heterostructure exhibits 84% photocatalytic efficiency (15 mg/L CIP concentration and 60 min irradiation period). Biochar and graphene have similar electrical properties, and the photogenerated holes are trapped by BiOBr reacting directly with CIP or H_2_O to form HO^−^ species. LaNiO_3_/TiO_2_ heterostructure has a granular morphology and was obtained by in situ sol–gel process. The S-scheme requires 180 min of irradiation in order to achieve 55% photocatalytic efficiency in a less concentrated CIP solution (10 mg/L). The higher energy consumption and lower photocatalytic efficiency reside on the low TiO_2_ activation under in visible spectra. However, under UV light (300 W), there is a significant increase of the photocatalytic efficiency, up to 90%, underlining the significance of matching the electronic structure of each component with the radiation source. Based on the potential difference of CB and VB in TiO_2_ and LaNiO_3_, they can form excellent S-scheme heterojunction.

The photocatalytic activity of type II heterostructure (UiO-66/CdIn_2_S_4_ [101]), p-n heterostructures (SnO_2_@ZnS [102]) and Schottky heterojunction (Ag/BiVO_4_/rGO [103]) was tested in 10 mg/L TCS solution. The optimum energy consumption corresponds to UiO-66/CdIn_2_S_4_, able to reach 92% photocatalytic efficiency after 180 min of irradiation with 150 W Vis light source. The UiO-66/CdIn_2_S_4_ type II heterostructure uses the combined advantages of better charge carrier’s channelization, high resistance to charge recombination (due to the favourable band alignment) and high specific 3D microflower-like morphology. Using Ag/BiVO_4_/rGO heterostructure obtained by a hydrothermal process, the TCS was completely removed after 120 min of irradiation with 300 W Vis light source. The high photocatalytic activity of Ag/BiVO_4_/rGO is the result of the three-way synergy mechanism, in which the photogenerated electrons from the CB of BiVO_4_ will benefit from the graphene transfer channels and the enhanced migration through Schottky Ag/BiVO_4_ barrier. SnO_2_@ZnS heterostructure with sheets morphology containing cubic ZnS and tetragonal SnO_2_ crystalline structures is able to attempt 40% photocatalytic efficiency after 120 min of irradiation with 500 W Vis light source. In this case, the energy consumption is higher due to interactions between TCS and photocatalyst surface. It was found that hydrated PhACs molecules are preferably adsorbed on the SnO_2_ surface rather than on the ZnS surface, which decreases the number of active sites.

The photocatalytic properties of Bi_7_O_9_I_3_/Bi_5_O_7_I [104] heterostructure were investigated using a 500 W Vis light source and 20 mg/L TCS aqueous solution. Bi_7_O_9_I_3_/Bi_5_O_7_I heterostructure with bone-stick-like morphology was obtained by in situ thermal treatment and exhibited 89.28% photocatalytic efficiency after 180 min of irradiation. The heterostructure contains two n-type semiconductors, where the photogenerated electrons are transferred from the CB of Bi_7_O_9_I_3_ to the CB of Bi_5_O_7_I and can be trapped by the molecular oxygen to form superoxidative radicals. The influence of the light spectra on the photocatalytic removal of highly concentrated TCS (50 mg/L) solution was studied on p-ZnIn_2_S_4_/rGO/n-g-C_3_N_4_ [105] heterostructure. The irradiation was done with a 20 W UV source and 2 W Vis source for 120 min. ZnIn_2_S_4_/rGO/g-C_3_N_4_ heterostructure with sheet-like morphology was obtained by hydrothermal process and exhibited similar photocatalytic efficiencies in UV (100%) and Vis (97%) radiation. It was concluded that it is possible to have ten times lower energy consumption using the same heterostructure, by optimizing the light spectra and intensity accordingly with the semiconductor components. The ternary system follows a Z-scheme charges transport, where the rGO accelerate the electron transfer and the band energy difference between g-C_3_N_4_ and ZnIn_2_S_4_ prolongs the charge lifetime and promotes electron-hole separation.

Three Z-scheme heterostructures (Ag/AgCl/BiVO_4_ [106], Ag/AgBr/ZnFe_2_O_4_ [107] and g-C_3_N_4_/TiO_2_ [108]) were employed to investigate CBZ (10 mg/L) photocatalytic removal under Vis light. Ag/AgCl/BiVO_4_ and Ag/AgBr/ZnFe_2_O_4_ were obtained by ultrasonication, and the photocatalytic experiments were done using similar parameters (93.38 W Vis light source and 240 min irradiation period). Three times more CBZ was removed using the same energy consumption by Ag/AgCl/BiVO_4_ heterostructure (70.6%), compared with Ag/AgBr/ZnFe_2_O_4_. The photocatalytic activity difference was attributed to the band energy positions, which are more close in Ag/AgCl/BiVO_4_ (−0.16 eV CB AgCl/0.33 eV CB BiVO_4_) heterostructure than that of Ag/AgBr/ZnFe_2_O_4_ (0.1 eV AgBr/−1.5 eV ZnFe_2_O_4_) heterostructure resulting in shorter transition time and lower recombination rate. g-C_3_N_4_/TiO_2_ heterostructure with sheets-like morphology obtained by calcination method shows better photocatalytic efficiency (99.77%), using lower radiation intensity (50 W) for a longer period (360 min). The high photocatalytic activity is attributed to g-C_3_N_4_, which has a high specific area and can enlarge the light absorbance spectra.

The SA photocatalytic removal in the presence of 300 W light source was studied using two WO_3_-based heterostructures obtained by one-pot (TiO_2_/WO_3_ [81]) and hydrothermal (WO_3_/Bi_2_WO_6_ [109]) processes. WO_3_/Bi_2_WO_6_ heterostructure with flower-like morphology was tested using 5 mg/L SA solution and Vis light source. After 360 min of irradiation, the WO_3_/Bi_2_WO_6_ heterostructure following a Z-scheme mechanism has reached 74.5% photocatalytic efficiency, due to the dramatic increases of visible light absorption induced by (1 1 0) facet of WO_3_. In this case, the active sites of the photocatalyst have been transferred from (0 1 0) facet of Bi_2_WO_6_ to (1 1 0) facet of WO_3_. The TiO_2_/WO_3_ heterostructure mechanism was previously presented in relation to MG dye. The photocatalytic evaluation was done under UV irradiation for 60 min, and the photocatalytic efficiency was 42%. The photocatalytic efficiency was correlated with the ten times higher SA concentration (50 mg/L) and five times shorter irradiation period. Compared with WO_3_/Bi_2_WO_6_, the TiO_2_/WO_3_ heterostructure has optimized energy consumption in correlation with the photocatalytic experimental conditions.

TiO_2_-NT’s@Ag-HA heterostructure [110] obtained by photoreduction was used to evaluate the influence of visible (100 W) and full spectrum (120 W) light on the photocatalytic activity toward 28 mg/L SA solution. As expected, due to the TiO_2_ content, after 240 min of irradiation, the photocatalytic efficiency under full spectrum (75%) was significantly higher than that under Vis spectrum (30%). The superior photocatalytic efficiency was attributed to the combined effect of a local surface plasmon resonance, induced by silver nanoparticles and the formation of additional levels in TiO_2_ band gap due to Ti^3+^ oxidation state at nanotubes surface. Bi_12_TiO_20_/g-C_3_N_4_ heterostructure [64] previously presented in relation to RhB dye was also tested for SA (10 mg/L) removal, in similar photocatalytic conditions (500 W Vis light source, 50 min irradiation period). However, the photocatalytic efficiency is considerably lower (50%) for SA compared with RhB (97%) due to the smaller SA adsorption on the heterostructure surface.

The removal of IBF (10 mg/L) was investigated using 60 W (Co_3_O_4_/BiOI [111]), 300 W (W_18_O_49_/g-C_3_N_4_ [112]) and 500 W (Fe_3_O_4_@MIL-53(Fe) [113]) Vis light sources intensities. Plates-like Co_3_O_4_/BiOI heterostructure obtained by solvothermal process exhibits 93.87% photocatalytic efficiency after 60 min of irradiation. The Co_3_O_4_/BiOI Z-scheme mechanism benefits from the improved separation of the photoexcited charge carriers, induced by the built-in electric field formed between the BiOI microspheres and the Co_3_O_4_ wormy epitaxy. In the presence of 300 W radiation source and 60 min irradiation period, the W_18_O_49_/g-C_3_N_4_ with sheets-like morphology exhibits 96.3% photocatalytic efficiency. Compared with Co_3_O_4_/BiOI, the photocatalytic efficiency improvement was moderate, considering that the radiation intensity was significantly higher. The same heterostructure was tested under NIR irradiation for 120 min, and the results indicate low photocatalytic efficiency (39.2%) due to the limited absorbance range. However, this value is promising, considering that most of the photocatalytic materials exhibit neither or very low photocatalytic activity under NIR irradiation. W_18_O_49_/g-C_3_N_4_ heterostructure follows a Z-scheme mechanism where the oxygen vacancies in W_18_O_49_ can lead to LSPR effect and induce the formation of hot electrons, which can significantly improve the amount of effective photogenerated electrons and provide unique hot electrons injection. Fe_3_O_4_@MIL-53(Fe) heterostructure with polyhedron particles morphology was obtained by calcinations, and the photocatalytic investigations were done using a 500 W Vis light source. The IBF was completely removed after 60 min irradiation period, due to high MIL-53(Fe) Vis absorbance induced by the spin-allowed d transition Fe^3+^ on Fe–O cluster. The energy consumption is higher compared with the previously presented studies. The calcination process was also used to produce Bi_2_O_3_-TiO_2_/carbon [114] heterostructure with S-scheme charges transport. The photocatalytic activity was evaluated using 20 mg/L IBF solution and 120 min irradiation with 300 W Vis light source. Bi_2_O_3_-TiO_2_/carbon heterostructure was able to completely remove the IBF due to the synergic role of Bi_2_O_3_ and TiO_2_ on photogenerated electrons and the tuneable charge carrier’s mobility through carbon channels.

High (100 mg/L) and low (2 mg/L) IBF concentrations were used to evaluate the photocatalytic activity of CdS–SnS–SnS_2_/rGO [115] and g-C_3_N_4_/TiO_2_/Fe_3_O_4_@SiO_2_ [116] heterostructures. CdS–SnS–SnS_2_/rGO containing hexagonal CdS, SnS_2_, and orthorhombic SnS was obtained by solvothermal process and exhibits sheets-like morphology. After 60 min of irradiation with 300 W visible light source, the photocatalytic efficiency achieved was 84.4%. The energy consumption is reasonable considering that the IBF concentration was 100 mg/L. CdS–SnS–SnS_2_/rGO benefits from a double heterojunction where rGO works as a mediator transfer for photogenerated electrons from CB of SnS to CB of SnS_2_ and finally to the CB of CdS. The lower IBF concentration (2 mg/L) was used to evaluate g-C_3_N_4_/TiO_2_/Fe_3_O_4_@SiO_2_ photocatalytic performance, during 15 min of irradiation with 64 W Vis source. g-C_3_N_4_/TiO_2_/Fe_3_O_4_@SiO_2_ heterostructure with sheets-like morphology was obtained by sol–gel method. The photocatalytic efficiency was 97%, and the energy consumption fits the photocatalytic parameters. Compared with CdS–SnS–SnS_2_/rGO photocatalytic parameters, the g-C_3_N_4_/TiO_2_/Fe_3_O_4_@SiO_2_ heterostructure uses not only 50 times lower IBF concentration but also 4.7 lower light intensity and 4 times shorter irradiation to exhibit good photocatalytic efficiency.

A 5 mg/L SMZ solution was employed to evaluate the photocatalytic activity of g-C_3_N_4_/TNTs [117] sheets-like and Pd-Bi_2_MoO_6_/g-C_3_N_4_ [118] flake-like heterostructures both following Z-scheme mechanisms. Pd–Bi_2_MoO_6_/g-C_3_N_4_ heterostructure is characterized by low energy consumption, being able to achieve 98.8% photocatalytic efficiency after 90 min of irradiation with 36 W Vis light source. In this heterostructure, Pd act as an electron mediator facilitating the charge migration. g-C_3_N_4_/TNTs heterostructure is able to induce total SMZ removal at high energy consumption, using a 450 W Vis source for 300 min irradiation period. The high photocatalytic efficiency is attributed to in situ transformations of titanate to anatase and rutile, leading to the formation of nanoscale “hot spots” and then subsequent charge transfer as well as to the large specific surface of TNTs skeleton. At higher SMZ (40 mg/L) concentration, the CuFe_2_O_4_/Ti_3_C_2_ [119] heterostructure with sheets-like morphology exhibits 70% photocatalytic efficiency after 60 min of irradiation with 300 W Vis light source. CuFe_2_O_4_/Ti_3_C_2_ follows a Z-scheme charges transport in which the Ti_3_C_2_ flakes serve as a shuttle and trap location for light-induced electrons.

To sum up, the most energy-efficient photocatalytic systems for PhACs removal correspond to Z-scheme ZnIn_2_S_4_/rGO/C_3_N_4_ heterostructure using 4 Wh to remove 48.5 mg/L of TCS and type II La(OH)_3_/BiOCl heterostructures requiring 5 Wh to remove 17 mg/L of TC. Higher energy consumption is attributed to p-n SnO_2_@ZnS heterostructures, which use 1000 Wh to remove 4.0 mg/L of TC.

## 4. Conclusions

The diversification of pollutant type and concentration in wastewater has underlined the importance of finding new alternatives to traditional treatment methods. AOPs, among others, are considered as a promising candidate to efficiently remove organic pollutants such as dyes of PhACs. The present minireview has considered several process parameters (radiation spectra, light intensity and irradiation period) and materials properties (crystallinity, morphology or charge transportation mechanism). However, there are also other parameters that were not the subject of this investigation but can have an important influence on the photocatalytic efficiency (specific surface, catalyst dosage, irradiance, pollutant type, etc.).

The presence of compatible crystalline structures between the heterostructure partners is a prerequisite to assure a good interfacial junction, which allows a facile charge carriers transport. The flower-like and sheets morphologies seem to host a larger number of active surface sites and, consequently, a higher photocatalytic activity. In order to obtain a balance between the energy consumption and photocatalytic efficiency, it is important to optimize the light spectra and intensity as well as the irradiance period according to the heterostructure type, pollutant molecule and concentrations. The total pollutant removal with the expense of high energy consumption will raise questions about the process sustainability and future large-scale implementation. It is possible to remove 85% of TC (20 mg/L) with low energy consumption (5 W Vis source, 60 min of irradiation) as well as 97.5% with higher energy consumption (300 W Vis, 50 min of irradiation). Sustainability is a key parameter to be considered when designing environmental treatment processes. AOPs based on photoactive heterostructures require chemically stable materials for long working periods and low energy consumption.

Dyes removal using photocatalytic heterostructures was evaluated for different molecules, and most of the results indicate high efficiencies. However, PhACs removal by photocatalytic processes still requires more experimental investigation on various molecules in order to have a detailed evaluation of the degradation mechanisms, including the by-products formation. The presence of by-products after the photocatalytic degradation of dyes and PhACs represents an important issue due to the hazard risk induced by their toxicity and environmental persistence. Consequently, the degradation mechanism of each pollutant molecule must be studied in relation to the photocatalytic parameters. The transfer from the laboratory investigations to large-scale applications will outline significant challenges in terms of economic costs and sustainability.

As perspectives, it will be recommended to use a uniform standardization regarding the photocatalytic activity experimental investigations. The scientific articles give several and often noncompatible parameters, which make the comparative investigation more difficult. For example, regarding the radiation parameters, most papers refer to light intensity (W), and other provides the irradiance (mW/cm^2^), which is more accurate. The photocatalytic efficiency improvement with few percentages is made by significant increase of the energy consumption instead of improving the intrinsic materials properties. Coupling photocatalysis with other techniques (adsorption, biodegradation, etc.) can be another pathway to follow for optimum energy consumption and organic pollutant removal.

## Figures and Tables

**Figure 1 nanomaterials-10-01766-f001:**
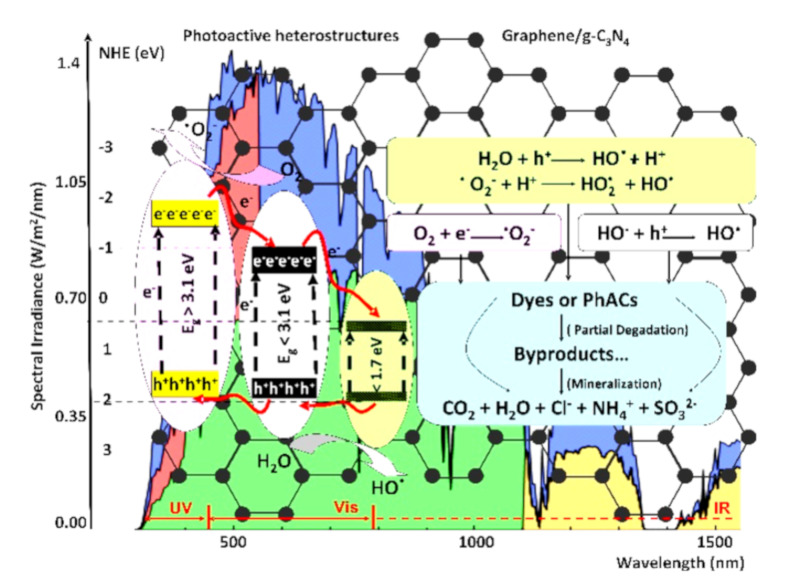
Photoactive heterostructures and the corresponding photocatalytic reactions.

**Figure 2 nanomaterials-10-01766-f002:**
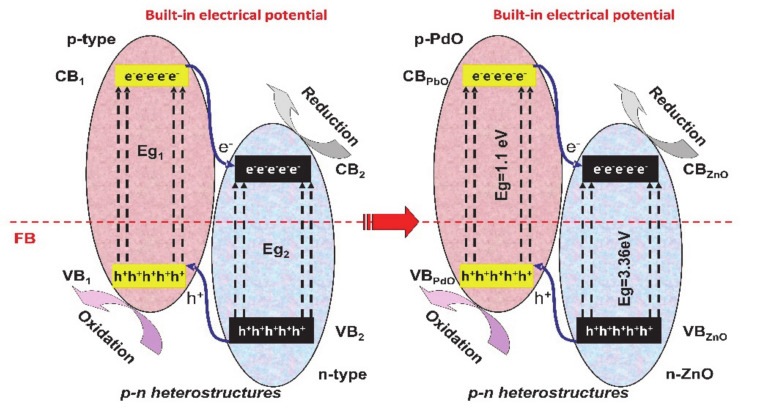
The p-n junction’s mechanism.

**Figure 3 nanomaterials-10-01766-f003:**
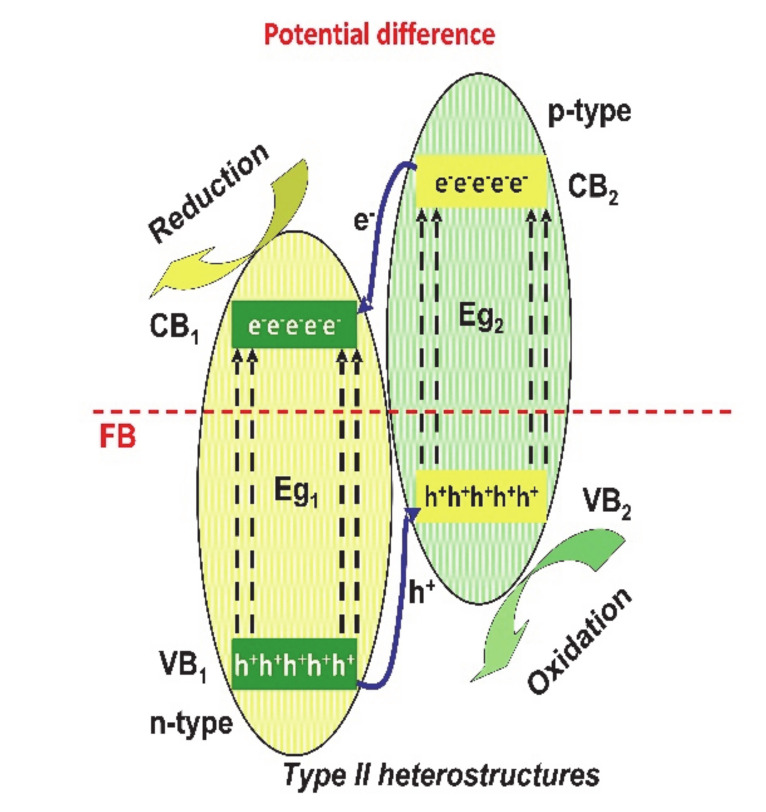
The type-II heterostructures mechanism.

**Figure 4 nanomaterials-10-01766-f004:**
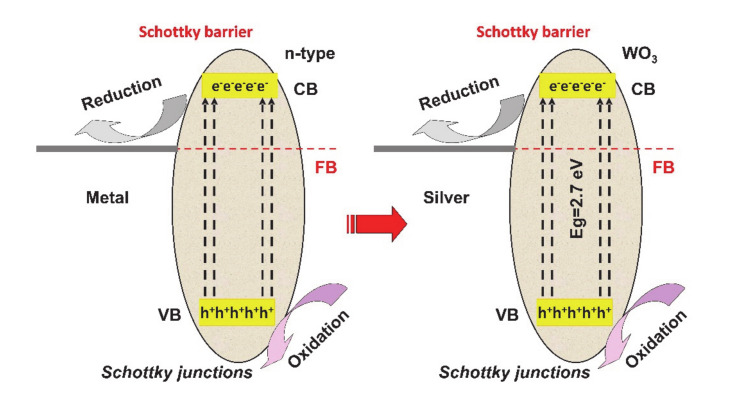
Schottky junction’s mechanism.

**Figure 5 nanomaterials-10-01766-f005:**
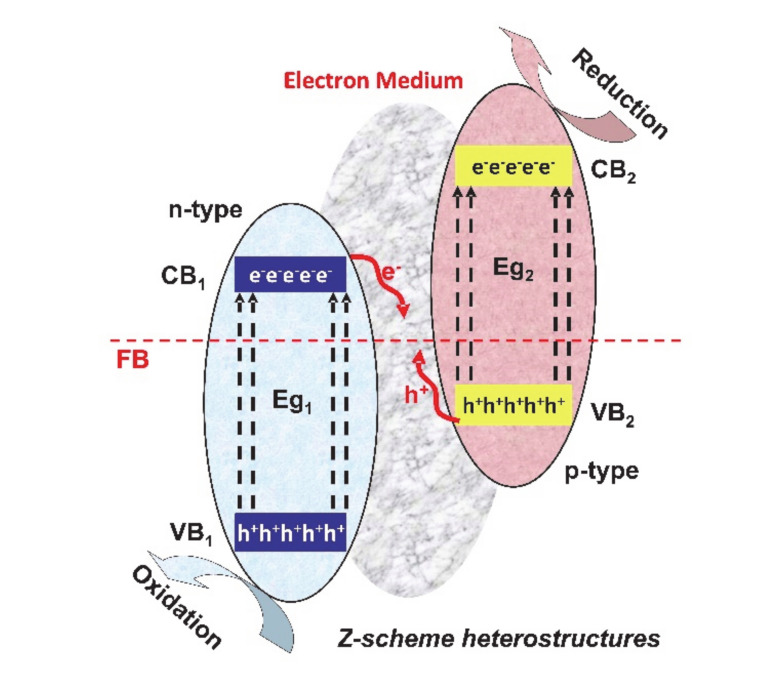
Z-scheme heterostructures for photocatalytic applications.

**Figure 6 nanomaterials-10-01766-f006:**
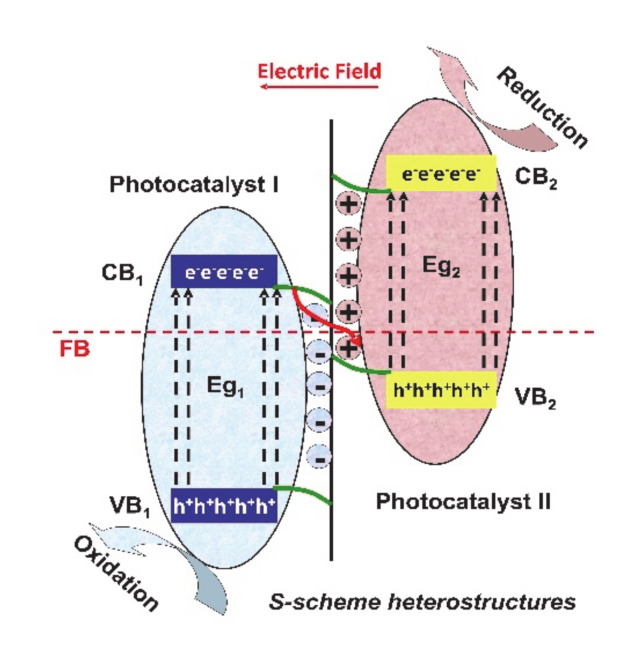
The mechanism of S-scheme heterostructures.

**Table 1 nanomaterials-10-01766-t001:** Recent representative studies on heterostructures photocatalytic application for dyes removal.

Heterostructure Composition	Synthesis Method	Morphology/Crystallinity	Pollutant and Concentration (mg/L)	Radiation Type, Intensity (*I*), Irradiation Time (*t*)	Efficiency and Rate Constant (min^−1^)	Ref.
Bi_12_TiO_20_/g-C_3_N_4_	Ultrasonication	Multilayer structure/cubic Bi_12_TiO_20_	RhB = 10 MO = 20	Vis *I* = 500 W *t* = 50 min	97% (RhB), 90% (MO)/0.0537 (RhB), 0.0328 (MO)	[64]
Bi_3.84_W_0.16_O_6.24_ (BWO)/g-C_3_N_4_	Ultrasonication	Sheets/cubic BWO	RhB = 10	Vis *I* = 100 W *t* = 50 min	99.8%/ 0.0562	[65]
Bi_2_MoO_6_/Bi_5_O_7_Br/TiO_2_	Solvothermal	Tube arrays/anatase TiO_2_, orthorhombic Bi_2_MoO_6_	RhB = 10 MO = 16 MB = 6.5	Vis *I* = 500 W *t* = 180 min	73.43% (RhB) 47.77% (MO) 93.81% (MB)/0.00742 (RhB) 0.00354 (MO) 0.00225 (MB)	[69]
Bi_2_MoO_6_/Fe_3_O	Solvothermal	Flower/orthorhombic Bi_2_MoO_6_	RhB = 20	Vis *I* = 350 W *t* = 120 min	99.5%/ 0.0364	[70]
NiO/BiOI	Solvothermal	Foam/crystalline NiO	RhB = 4.8	Vis *I* = 300 W *t* = 60 min	90%/ 0.0572	[60]
TiO_2_/g-C_3_N_4_	Hydrothermal	2D sheet/pristine 2D-TiO_2_	RhB = 10	Vis *I* = 500 W *t* = 60 min	85%/ 0.03	[66]
WS_2_/BiOBr	Hydrothermal	Plates/tetragonal BiOBr	RhB = 20	Vis *I* = 500 W *t* = 100 min	95%/ np *	[68]
g-C_3_N_4_/ZnO	Hydrothermal	Rod/hexagonal wurtzite ZnO	RhB = 10 MB = 10	Vis *I* = 300 W *t* = 70 min	98% (MB), 98.5% (RhB)/ np	[67]
BiPO_4−x_/B_2_S_3_	Hydrothermal	Sheet/monoclinic BiPO_4_ and orthorhombic Bi_2_S_3_	MB = 5	Vis *I* = 300 W *t* = 360 min	98%/ 0.0222	[72]
MnFe_2_O_4_/rGO	Coprecipitation	Spherical/cubic MnFe_2_O_4_	MB = 10	UV *I* = 40 W *t* = 60 min	97%/ 0.0589	[73]
ZnAl_2_O_4_/Bi_2_MoO_6_	Coprecipitation	Sheet/koechlinite Bi_2_MoO_6_ and gahnite ZnAl_2_O_4_	MB = 30	UV *I* = 100 W *t* = 180 min	86.36%/ 0.638	[75]
Ag/hybridized 1T-2H MoS_2_/TiO_2_	Chemical reduction	Flower/anatase TiO_2_	MB = 20	UV *I* = 235 W *t* = 60 min	96.8%/ 0.0539	[74]
Ta_3_B_2_@Ta_2_O_5_	In situ	Powder/crystalline Ta_3_B_2_ and Ta_2_O_5_	MB = 50	Vis *I* = 500 W *t* = 180 min	80%/ np	[71]
CuO–TiO_2_	Ultrasonication	Fiber/anatase TiO_2_ and monoclinic CuO	MB = 1	UVc, Vis *I*_UVc_ = 96 W *I*_Vis_ = 250 W *t*_UVc_ = 30 min *t*_Vis_ = 240 min	99% (UVc) 98% (Vis)/ 0.135 (UVc) 0.015 (Vis)	[76]
WO_3_/g-C_3_N_4_	Polymerization	Sheet/crystalline WO_3_/g-C_3_N_4_	MO = 10	Vis *I* = 300 W *t* = 120 min	93%/ 0.0213	[79]
LaNiO_3_/TiO_2_	Sol–gel	Particles/perovskite LaNiO_3_, anatase and rutile TiO_2_	MO = 10 MO = 20	Vis *I* = 300 W *t* = 150 min	100% (10 mg/L) 92% (20 mg/L)/ np	[77]
ZnFe_2_O_4_/SnS_2_	Solvothermal	Particles/crystalline ZnFe_2_O_4_ and SnS_2_	MO = 50	Vis *I* = 300 W *t* = 20 min	99%/ 0.214	[78]
Ag_2_Mo_1−x_W_x_O_4_	Microwave-assisted hydrothermal	Rod/cubic Ag_2_MoO_4_, orthorhombic Ag_2_WO_4_	MO = 5	UVc *I* = 90 W *t* = 140 min	45%/ 0.0058	[80]
TiO_2_/WO_3_	One pot	Hollow sphere/anatase TiO_2_ and monoclinic WO_3_	MG = 50	Vis *I* = 300 W *t* = 60 min	98%/ 0.0746	[81]
La_2_CuO_4_-decorated ZnO	In situ extraction	Particles/crystalline ZnO, orthorhombic La_2_CuO_4_	MG = 25	Vis *I* = 125 W *t* = 120 min	91%/ 0.063	[83]
MgFe_2_O_4_/Bi_2_MoO_6_	Hydrothermal	Plates/crystalline Bi_2_MoO_6_ and MgFe_2_O_4_	MG = 20	Vis *I* = 300 W *t* = 120 min	97%/ 0.0113	[84]
CdS@ZnS@ZnO	Hydrothermal	Spherical/cubic ZnS, hexagonal CdS and ZnO	MG = 50	UV, Vis *I*_UV_ = 125 W *I*_Vis_ = 400 W *t*_UV_ = 30 min *t*_Vis_ = 180 min	95% (UV) 65% (Vis)/ np	[82]

* not provided.

**Table 2 nanomaterials-10-01766-t002:** Recent representative studies on heterostructures photocatalytic application for pharmaceutical active compounds removal.

Heterostructure Composition	Synthesis Method	Morphology/Crystallinity	Pollutant and Concentration (mg/L)	Radiation Type, Intensity (*I*), Irradiation Time (*t*)	Efficiency and Rate Constant (min^−1^)	Ref.
BN/B-doped-g-C_3_N_4_	In situ growth	Sheet/hexagonal BN	TC = 10	Vis *I* = 300 W *t* = 60 min	88.1%/ 0.034	[91]
WO_3_/g-C_3_N_4_	Polymerization	Sheet/crystalline WO_3_/g-C_3_N_4_	TC = 10	Vis *I* = 300 W *t* = 180 min	97%/ np *	[79]
Ag_3_PO_4_/Co_3_(PO_4_)_2_/g-C_3_N_4_	Precipitation	3D flower/crystalline Co_3_(PO_4_)_2_, g-C_3_N_4_ and Ag_3_PO_4_	TC = 10	Vis *I* = 300 W *t* = 120 min	88%/ 0.0159	[93]
g-C_3_N_4_-decorated ZrO_2−x_	Anodic oxidation and PVD	Tube/tetragonal and monoclinic zirconia	TC = 10	Vis *I* = 300 W *t* = 60 min	90.6%/ 0.0474	[92]
ZnIn_2_S_4_/BiPO_4_	Hydrothermal	Flower/monoclinic BiPO_4_	TC = 40	Vis *I* = 300 W *t* = 90 min	84%/ 0.0201	[98]
AgI/Bi_2_MoO_6_/AgBi(MoO_4_)_2_	Hydrothermal	Sheets/crystalline AgI, Bi_2_MoO_6_ and AgBi(MoO_4_)_2_	TC = 5	Vis *I* = 400 W *t* = 90 min	91.9%/ 0.0097	[99]
MoS_2_/g-C_3_N_4_/Bi_24_O_31_Cl_10_	Calcination	Sheet/monoclinic Bi_24_O_31_Cl_10_ and MoS_2_	TC = 20	Vis *I* = 300 W *t* = 50 min	97.5%/ 0.0642	[94]
CuBi_2_O_4_/Bi_2_WO_6_	Hydrothermal	Pseudo-sphere/tetragonal CuBi_2_O_4_ and orthorhombic Bi_2_WO_6_	TC = 20	Vis *I* = 300 W *t* = 120 min	93%/ 0.0286	[95]
La(OH)_3_/BiOCl	Microwave	Sheet/crystalline BiOCl	TC = 20	Vis *I* = 5 W *t* = 60 min	85%/ 0.037	[96]
WS_2_/BiOBr	Hydrothermal	Plates/tetragonal BiOBr	TC = 20 CIP = 20	Vis *I* = 500 W *t* = 100 min	96% (TC) 92% (CIP)/ np (TC) 0.01708 (CIP)	[68]
BiOCl/CQDs/rGO	Hydrothermal	Sheet/tetragonal BiOCl	CIP = 20	Vis *I* = 300 W *t* = 100 min	87%/ 0.0146	[97]
LaNiO_3_/TiO_2_	In situ sol–gel	Granular/anatase TiO_2_ and perovskite LaNiO_3_	CIP = 10	UV, Vis *I*_UV, Vis_ = 300 W *t* = 180 min	90% (UV) 55% (Vis)/ np	[77]
PVPbiochar@ZnF_2_O_4_/BiOBr	Solvothermal	Sheet/tetragonal BiOBr, spinel ZnFe_2_O_4_	CIP = 15	Vis *I* = 300 W *t* = 60 min	84%/ np	[100]
UiO-66/CdIn_2_S_4_	Solvothermal	3D flower/pristine CIS	TCS = 10	Vis *I* = 150 W *t* = 180 min	92%/ 0.0094	[101]
Ag/BiVO_4_/rGO	Hydrothermal	Irregular Particles/monoclinic BiVO_4_	TCS = 10	Vis *I* = 300 W *t* = 120 min	100%/ np	[103]
SnO_2_@ZnS	Hydrothermal	Sheet/cubic ZnS, tetragonal SnO_2_	TCS = 10	Vis *I* = 500 W *t* = 120 min	40%/ 0.0033	[102]
Bi_7_O_9_I_3_/Bi_5_O_7_I	Calcination	Bone-stick/crystalline Bi_7_O_9_I_3_ and Bi_5_O_7_I	TCS = 20	Vis *I* = 500 W *t* = 180 min	89.28%/ 0.0168	[104]
p-ZnIn_2_S_4_/rGO/n-g-C_3_N_4_	Hydrothermal	Sheet/crystalline ZnIn_2_S_4_	TCS = 50	UV, Vis *I*_UV_ = 20 W *I*_Vis_ = 2 W *t* = 120 min	100% (UV) 97% (Vis)/ np	[105]
Ag/AgCl/BiVO_4_	Ultrasonication	Octahedral particle/monoclinic BiVO4, crystalline AgCl and Ag	CBZ = 10	Vis *I* = 93.38 W *t* = 240 min	70.6%/ np	[106]
g-C_3_N_4_/TiO_2_	Calcination	Sheet/crystalline g-C_3_N_4_, anatase TiO_2_	CBZ = 10	Vis *I* = 50 W *t* = 360 min	99.77%/ 0.1796	[108]
Ag/AgBr/ZnFe_2_O_4_	Ultrasonication	Spherical/cubic AgBr and ZnFe_2_O_4_	CBZ = 10	Vis *I* = 93.38 W *t* = 240 min	22.7%/ np	[107]
Bi_12_TiO_20_/g-C_3_N_4_	Ultrasonication	Spherical/cubic Bi_12_TiO_20_	SA = 10	Vis *I* = 500 W *t* = 50 min	50%/ np	[64]
WO_3_/Bi_2_WO_6_	Hydrothermal	Flower/orthorhombic Bi_2_WO_6_	SA = 5	Vis *I* = 300 W *t* = 360 min	74.5%/ 0.00435	[109]
TiO_2_-NT’s@Ag-HA	Photoreduction	Tubes/anatase TiO_2_	SA = 28	Full Spectrum (FS), Vis *I*_FS_ = 120 W *I*_Vis_ = 100 W *t* = 240 min	75% (FS) 30% (Vis)/ 0.00581 (FS) 0.00129 (Vis)	[110]
TiO_2_/WO_3_	One pot	Hollow sphere/anatase TiO_2_ and monoclinic WO_3_	SA = 50	UV *I* = 300 W *t* = 60 min	42%/ np	[81]
CdS-SnS-SnS_2_/rGO	Solvothermal	Sheet/hexagonal CdS, SnS_2_ and orthorhombic SnS	IBF = 100	Vis *I* = 300 W *t* = 60 min	84.4%/ 0.0257	[115]
Bi_2_O_3_-TiO_2_/carbon	Calcination	Particle/anatase and rutile TiO_2_	IBF = 20	Vis *I* = 300 W *t* = 120 min	100%/ 0.0290	[114]
W_18_O_49_/g-C_3_N_4_	Hydrothermal	Sheet/monoclinic W_18_O_49_	IBF = 10	Vis, NIR *I*_Vis, NIR_ = 300 W *t*_Vis_ = 60 min *t*_NIR_ = 120 min	96.3% (Vis) 39.2% (NIR)/0.0464 (Vis) 0.0027 (NIR)	[112]
Fe_3_O_4_@MIL-53(Fe)	Calcination	Particles with polyhedron structure/crystalline Fe_3_O_4_	IBF = 10	Vis *I* = 500 W *t* = 60 min	99%/ 0.0471	[113]
Co_3_O_4_/BiOI	Solvothermal	Plates/crystalline Co_3_O_4_, Tetragonal BiOI	IBF = 10	Vis *I* = 60 W *t* = 60 min	93.87%/ 0.0945	[111]
g-C_3_N_4_/TiO_2_/Fe_3_O_4_@SiO_2_	Sol–gel	Sheet/standard magnetite, anatase TiO_2_	IBF = 2	Vis *I* = 64 W *t* = 15 min	97%/ np	[116]
g-C_3_N_4_/TNTs	Hydrothermal	Sheet/anatase and rutile TiO_2_	SMZ = 5	Vis *I* = 450 W *t* = 300 min	100%/ 0.0193	[117]
Pd-Bi_2_MoO_6_/g-C_3_N_4_	Precipitation	Flake/crystalline Bi_2_MoO_6_ and g-C_3_N_4_	SMZ = 5	Vis *I* = 36 W *t* = 90 min	98.8%/ 0.0440	[118]
CuFe_2_O_4_/Ti_3_C_2_	Hydrothermal	Sheet/spinel CuFe_2_O_4_	SMZ = 40	Vis *I* = 300 W *t* = 60 min	70%/ 0.0128	[119]

* not provided.

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
