# Peer review of "The Influence of Photoactive Heterostructures on the Photocatalytic Removal of Dyes and Pharmaceutical Active Compounds: A Mini-Review"

_nanomaterials, 2020, doi:10.3390/nano10091766_

Round 1

Reviewer 1 Report

The authors placed efforts to revise their manuscript. However, the key problem is the interpretation of what a Review should be. This referee still sees poor interpretation and attempt of synthesis in the wide number of papers analysed. The manuscript still represents a short summary of papers rather than a discussion on the effect of the selected structures on performance. Therefore, considering also the choice of model Molecules which are not applicatively relevant, the practical importance is very limited. I still do not recommend the manuscript for publication.

Author Response

Dear Reviewer,

We express our gratitude to your work and guidance that was helping us to improve the quality of this manuscript.

We have considered all your comments and suggestion in the new revised form of the manuscript. The changes where highlighted in red.

The authors placed efforts to revise their manuscript. However, the key problem is the interpretation of what a Review should be. This referee still sees poor interpretation and attempt of synthesis in the wide number of papers analysed. The manuscript still represents a short summary of papers rather than a discussion on the effect of the selected structures on performance.

A1. The manuscript was improved in order to include a balanced analysis of several factors that may influence the energy consumption. There are also other factors which were not included as the space is limited. In this sense the authors have underline the statement which mention “Many other papers which are not included here have the potential to contain highly innovative work.” as well as the fact that “The lack of standardization regarding the photocatalysis experimental parameters has as consequence the presence of various scientific papers containing data difficult to compare. “

The following changes were done to improve the manuscript:

Lines: 13-15

Lines: 52-53

Lines: 60-62

Lines: 138-141

Lines: 145-154

Lines: 278-284

Lines: 289-292

Lines: 495-499

Therefore, considering also the choice of model Molecules which are not applicatively relevant, the practical importance is very limited. I still do not recommend the manuscript for publication.

A2. The model molecules were chosen based on the statistical number of publications in relevant journals (Applied Catalysis A and B, Journal of Hazardous Materials, Environmental Pollution, etc.) as well as the molecules reference data sheet considered in this field. The dyes molecules (RhB, MB, MO and MG) exhibit good chemical stability toward traditional and modern wastewater treatments due to aromatic structures. Recent studies have shown that the PhACs concentrations (TC, TCS, CIP, CBZ, SA, IBF and SMZ) in householders and hospital wastewater have increase significantly. The use of AOPs open new possibility of adopting sustainable technologies to remove chemically stable organic pollutants.

We have outlined into the manuscript that there are many other parameters and pollutants which may play an important role in this field. We can’t consider all of them due to the space limitation.

Lines: 54-55

Lines: 506-508

Lines: 523-529

Thank you!

Reviewer 2 Report

The revised review solved most of my questions, and the quality of the manuscript was improved as well. It can be published in Nanomaterials after the following issues have been well addressed.

  1. Line 13-14 and Line 52, “The present mini-review represents a synthesis of the some recent achievement on the implementation and …” what does “a synthesis” mean? I cannot understand this point.
  2. Line 145, please double check the photocatalyst in Ref. 65, I was confused on this. I thought it should be” Bi3.84W0.16O6.24” rather than “B3.84W0.16O6.24” . "Bi" is different from "B", and theoretically it should be "Bi" since the authors use Bismuth(III) nitrate as the precusor. 
  3. Please check the spelling of the words in the whole manuscript. I can still see some typos. e.g. Line 508 “bi-product” should be “by-product"  
  4. In the conclusion part, please highlight the toxicity of the “by-product” or intermediates should also be investigated when the dyes and PhACs were not totally degraded.
  5. All the chemical formula in this manuscript was not correct. The numbers in the chemical formula should be subscripted. Please revise them accordingly.

Author Response

Dear Reviewer,

We express our gratitude to your work and guidance that was helping us to improve the quality of this manuscript.

We have considered all your comments and suggestion in the new revised form of the manuscript. The changes where highlighted in red.

The revised review solved most of my questions, and the quality of the manuscript was improved as well. It can be published in Nanomaterials after the following issues have been well addressed.

Q1. Line 13-14 and Line 52, “The present mini-review represents a synthesis of the some recent achievement on the implementation and …” what does “a synthesis” mean? I cannot understand this point.

A1. We have corrected the two sentences in order to make them clear for the readers.

Lines 13-15 and 52-53

Q2. Line 145, please double check the photocatalyst in Ref. 65, I was confused on this. I thought it should be” Bi3.84W0.16O6.24” rather than “B3.84W0.16O6.24” . "Bi" is different from "B", and theoretically it should be "Bi" since the authors use Bismuth(III) nitrate as the precusor. 

A2. Thank you for the observation. We have corrected this mistake.

Table 1

Lines 145-154

Q3. Please check the spelling of the words in the whole manuscript. I can still see some typos. e.g. Line 508 “bi-product” should be “by-product" 

A3. The manuscript was check for typo and grammar mistakes.

Line 525-526 was corrected as well.

Q4. In the conclusion part, please highlight the toxicity of the “by-product” or intermediates should also be investigated when the dyes and PhACs were not totally degraded.

A4. Thank you for the suggestion. We introduced additional sentences to outline this aspect.

Lines 526-529

Q5. All the chemical formula in this manuscript was not correct. The numbers in the chemical formula should be subscripted. Please revise them accordingly.

A5. Due to the word files conversions some of the subscript were cancelled. We have corrected all the chemical formula.

Thank you!

Reviewer 3 Report

Review of The influence of photoactive heterostructures on the photocatalytic removal of dyes and pharmaceutical active compounds: A mini-review by Alexandru Enesca and Luminita Andronic.

The authors have addressed the main concerns of my earlier review.
The Abstract gives a clear statement of purpose.
The Conclusions provide a synthesis of the knowledge and clear recommendations.
The table format has been improved.

There are many details in this review which will be useful to other researchers.

One significant concern remains. There are errors in the English grammar and style that significantly detract from the quality of the paper and must be corrected before publication.

Author Response

Dear Reviewer,

We express our gratitude to your work and guidance that was helping us to improve the quality of this manuscript.

We have considered all your comments and suggestion in the new revised form of the manuscript. The changes where highlighted in red.

Review of The influence of photoactive heterostructures on the photocatalytic removal of dyes and pharmaceutical active compounds: A mini-review by Alexandru Enesca and Luminita Andronic.

The authors have addressed the main concerns of my earlier review.
The Abstract gives a clear statement of purpose.
The Conclusions provide a synthesis of the knowledge and clear recommendations.
The table format has been improved.

There are many details in this review which will be useful to other researchers.

Q1. One significant concern remains. There are errors in the English grammar and style that significantly detract from the quality of the paper and must be corrected before publication.

A1. Thank you for the observation. Due to the extensive changed made on the manuscript some grammar and typo mistakes were present. We have done the necessary corrections through the entire manuscript.

Thank you!

Reviewer 4 Report

The manuscript reviews the relations between the synthesis, the structural properties and the photocatalytical efficiency of heterostructures when applied in water purification processes under UV-Vis irradiation. The manuscript offers a good and updated overview on the different heterostructures produced and used firstly for the removal of dyes and pharmaceutical active compounds, indeed the most of the papers taken as references have been published in the last three years. I consider the paper suitable for publication, but before this, some major revisions should be done.

Entire manuscript:

  • The manuscript needs a check for grammar and typo errors. Same sentences should be rewritten.
  • The numbering of the figures and the content of the captions should be properly checked.

Introduction:

  • Some references seem inappropriate, for example I don’t understand the use of the references 2, 3 and 6.
  • For general descriptions related to well assessed processes and issues also the reference to book is kindly encouraged.
  • Line 40: what do the authors mean with "breakthrough heterostructures"?
  • Lines 60-62: please rewrite, it is quite confused.

Paragraph 2:

  • Line 64: the words "chemical reaction" should be changed with the word "process"
  • Along the quite specific references 31-38 also some reviews or books should be listed.
  • The Figure 1 should be improved and simplified.
  • Subsection 3.1, line 135: the word "radials" should be changed with the word "groups".
  • Subsection 3.1: add a proper introduction of the Table 1, specifying its contents.
  • Section 3.1, line 144: something is missing, please check the sentence.
  • Section 3.2: add a proper introduction of the Table 2, specifying its contents.
  • Subsection 3.2, lines 378-379, please rewrite.
  • Subsection 3.2, line 406: please check the sentence, the photocatalytical removal of SA is tested not its photocatalytical activity.
  • Both subsections 3.1 and 3.2 are the result of the resumes of single articles and an overall evaluation of the most important characteristics expected for a best performing heterostructure in dye or PhACs removal is missing. I suggest to add at the end of each subsection, something like: "to sum up…" or to move part of the conclusions to the relative subsections.

Conclusions

  • Some parts of the conclusions should be moved to the substections 3.1 and 3.2 as suggested in the previous comment.
  • The authors should add some aspects related to the costs of productions of materials and their production at large scale for a real and practical future application of these materials in the water streams treatment.

Author Response

Dear Reviewer,

We express our gratitude to your work and guidance that was helping us to improve the quality of this manuscript.

We have considered all your comments and suggestion in the new revised form of the manuscript. The changes where highlighted in red.

The manuscript reviews the relations between the synthesis, the structural properties and the photocatalytical efficiency of heterostructures when applied in water purification processes under UV-Vis irradiation. The manuscript offers a good and updated overview on the different heterostructures produced and used firstly for the removal of dyes and pharmaceutical active compounds, indeed the most of the papers taken as references have been published in the last three years. I consider the paper suitable for publication, but before this, some major revisions should be done.

Q1. The manuscript needs a check for grammar and typo errors. Same sentences should be rewritten.

A1. Thank you for the observation. Due to the extensive changed made on the manuscript some grammar and typo mistakes were present. We have done the necessary corrections through the entire manuscript.

Q2. The numbering of the figures and the content of the captions should be properly checked.

A2. The figures captions and numbering were corrected.

Figure 2

Figure 5

Figure 6

Line 122

Q3. Some references seem inappropriate, for example I don’t understand the use of the references 2, 3 and 6.

A3. We have replaced the references 2, 3 and 6 with others more appropriate.

Reference 2 “Singh, R.P.; Mishra, S.; Das, A.P. Synthetic microfibers: Pollution toxicity and remediation. Chemosphere 2020, 257, 127199.” was replaced with “Gautam, K.; Anbumani, S. Ecotoxicological effects of organic micro-pollutants on the environment. In Current Developments in Biotechnology and Bioengineering, 1st ed.; Varjani, S., Pandey, A., Tyagi, R. D., Ngo, H. H., Larroche, C., Eds.; Elsevier: New York, USA, 2020; pp. 481-501.”

Reference 3 “Al-Kandari, H.; Younes, N.; Al-Jamal, O.; Zakaria, Z.Z.; Najjar, H.; Alserr, F.; Pintus, G.; Al-Asmakh, M.A.; Abdullah, A.M.; Nasrallah, G.K. Ecotoxicological assessment of thermally- and hydrogen-reduced graphene oxide/TiO2 photocatalytic nanocomposites using zebrafish embryo model. Nanomaterials 2019, 9, 488.” was replaced with “3.           Duarte, R. M.; Matos, J. T.; Senesi, N. Organic pollutants in soils. In Soil Pollutions, 1st ed.; Duarte, A. C., Cachada, A., Rocha-Santos, T, Eds.; Elsevier: New York, USA, 2018; pp. 103-126.”

Reference 6 “Pratiti, R.; Vadala, D.; Kalynch, Z.; Sud, P. Health effects of household air pollution related to biomass cook stoves in resource limited countries and its mitigation by improved cookstoves. Environ. Res. 2020, 186, 109574.” was replaced with “Libralato, G.; Lofrano, G.; Siciliano, A.; Gambino, E.; Boccia, G.; Federica, C.; Francesco, A.; Galdiero, E.; Gesuele, R.; Guida, M. Toxicity assessment of wastewater after advanced oxidation processes for emerging contaminants’ degradation. In Visible Light Active Structured Photocatalysts for the Removal of Emerging Contaminants, 1st ed.; Sacco, O., Vaiano, V., Eds.; Elsevier: New York, USA, 2020; pp. 195-211.”

Q4. For general descriptions related to well assessed processes and issues also the reference to book is kindly encouraged.

A4. We have inserted 3 books as references:

  • Current Developments in Biotechnology and Bioengineering, 1st ed.; Varjani, S., Pandey, A., Tyagi, R. D., Ngo, H. H., Larroche, C., Eds.; Elsevier: New York, USA, 2020.
  • Soil Pollutions, 1st ed.; Duarte, A. C., Cachada, A., Rocha-Santos, T, Eds.; Elsevier: New York, USA, 2018.
  • Visible Light Active Structured Photocatalysts for the Removal of Emerging Contaminants, 1st ed.; Sacco, O., Vaiano, V., Eds.; Elsevier: New York, USA, 2020.

Q5. Line 40: what do the authors mean with "breakthrough heterostructures"?

A5. The term "breakthrough heterostructures" is used for the new developed heterostructures able to extend the light absorption range up to NIR in order to generate extra charge carriers involved in the formation of (super)oxidative species.

Q6. Lines 60-62: please rewrite, it is quite confused.

A6. The phrase was rewrite as suggested.

Lines 60-62

Q7. Line 64: the words "chemical reaction" should be changed with the word "process"

A7. We have replaced the word “reaction” with “process”.

Line 64

Q8. Along the quite specific references 31-38 also some reviews or books should be listed.

A8. Thank you for the suggestion. Two books and two reviews were inserted.

Books:

  • Handbook of Smart Photocatalytic Materials, 1st ed.; Hussain, C. M., Mishra, A.K., Eds.; Elsevier: New York, USA, 2020; pp. 39-57.
  • Nanomaterials Applications for Environmental Matrices, 1st ed.; Nascimento, R. F., Ferreira, O. P., De Paula, A. J., Neto, V. O., Eds.; Elsevier: New York, USA, 2019; pp. 449-488.

Reviews:

  • Zhang, X.; Teng, S.Y.; Loy, A.C.M.; How, B.S.; Leong, W.D.; Tao, X. Transition Metal Dichalcogenides for the Application of Pollution Reduction: A Review. Nanomaterials 2020, 10, 1012.
  • Song, B.; Zeng, Z.; Zeng, G.; Gong, J.; Tang, X. Powerful combination of g-C3N4 and LDHs for enhanced photocatalytic performance: A review of strategy, synthesis, and applications. Adv. Colloid Interfac. 2019, 272, 101999.

Q9. The Figure 1 should be improved and simplified.

A9. Figure 1 was improved and simplified as suggested.

Q10. Subsection 3.1, line 135: the word "radials" should be changed with the word "groups".

A10. We have replaced the word “radials” was replaced with the word “groups”.

Line 134

Q11. Subsection 3.1: add a proper introduction of the Table 1, specifying its contents.

A11. We have inserted a proper introduction of the Table 1.

Lines 138-141

Q12. Section 3.1, line 144: something is missing, please check the sentence.

A12. We have corrected the sentence.

Lines 145-147

Q13. Section 3.2: add a proper introduction of the Table 2, specifying its contents.

A13. We have inserted a proper introduction of the Table 2.

Lines 289-292

Q14. Subsection 3.2, lines 378-379, please rewrite.

A14. We have rewritten the sentence.

Lines 391-392

Q15. Subsection 3.2, line 406: please check the sentence, the photocatalytical removal of SA is tested not its photocatalytical activity.

A15. True. The mistake was corrected. Thank you!

Line 419

Q16. Both subsections 3.1 and 3.2 are the result of the resumes of single articles and an overall evaluation of the most important characteristics expected for a best performing heterostructure in dye or PhACs removal is missing. I suggest to add at the end of each subsection, something like: "to sum up…" or to move part of the conclusions to the relative subsections.

Q17. Some parts of the conclusions should be moved to the substections 3.1 and 3.2 as suggested in the previous comment.

A16&17. We have added at each subsection a short conclusion. We made changes in the conclusion subsections as well.

Lines 278-284

Lines 495-499

Q18. The authors should add some aspects related to the costs of productions of materials and their production at large scale for a real and practical future application of these materials in the water streams treatment.

A18. Thank you for the suggestions. We have already tried to make an economic assessment, without success. The energy and materials prices have significant fluctuation based on the geographical area, suppliers, legislation, etc. Additionally, large scale implementation must consider the economic support of regional entities based on dedicated positive environmental impact funds. These aspects exceed our area of expertise.

Thank you!

Round 2

Reviewer 1 Report

The main criticisms remain after this new revision. The Review lacks of wide breath and general viewpoint. It still represents a list of papers with poor connection among each others.

Sorry to confirm the opinion against publication.

Reviewer 4 Report

The authors addressed all the comments and the questions I formulated during the first round of revision improving the manuscript accordingly. The revised version of the manuscript is more complete and readable. The manuscript is suitable for publication to me in the present form.

This manuscript is a resubmission of an earlier submission. The following is a list of the peer review reports and author responses from that submission.

Round 1

Reviewer 1 Report

Review of The influence of photoactive heterostructures on the photocatalytic removal of dyes and pharmaceutical active compounds: A mini-review by Alexandru Enesca and Luminita Andronic.

This is a short review which could be significantly improved by revision. Recommend major revision.

Please go through the English grammar carefully, it might be very worthwhile to find help from an expert. There are mistakes in every paragraph, practically every sentence, that take away from the professional level.

In the introduction, please do a better job of arguing for why this review is necessary. There are many other review papers. Some of them should be referenced. Are they inadequate for some reason? Is there a gap in our knowledge or recent developments, that justify the current work?

I would like to see more analysis and synthesis (not in the chemical sense but rather 'the combination of components or elements to form a connected whole'). Right now the article is primarily descriptive but it does not give an overview, a context, recommendations for next steps, analysis of what works and what doesn't and why. For example line 49, 'The paper is focused on various methods of enhancing the heterostructure photocatalytic properties by optimizing parameters such as synthesis methods, composition, crystallinityt, morphology and pollutant concentration and light irradiation.' That is an ambitious list. If this is the focus of the paper, then I would like the authors to make conclusioons - how do these six parameters affect heterostructure photocatalytic properties? How are heterostructure photocatalytic properties defined, can you put a number or numbers on it?

Line 65, repeats what you often hear, that there is mineralization and degradation to CO2 and H2O. I feel this is inadequate because many photocatalytic installations work slowly and therefore mineralization is very incomplete. They make a huge range of byproducts that end up being the final products because of the limitations of the system. 'Mineralization' seems like a stage in the development that is never reached. More mention should be made of intermediate degradatioon products instead of sweeping them under the rug.

In Figure 1, we see the reactions H2O + h+ --> OH + H+ and O2 + e- --> superoxide.
What measurements support the conclusion that these are the main reactions? Can e- and h+ react directly with organics adsorbed on surfaces?

Figure 2 should be improved. It is difficult to see the arrows of e- and h+ transfer (heads are too small). Make it clear which arrows represent light.

The fourth column of Tables 1 and 2 is a mess. No less than five parameters are squeezed into a single column. This makes it hard to read and compare between different entries. Please revise these tables.

In the Conclusions line 255, 'As perspective, the heterostructure optimizations must consider a balance betyween the energy consumption and photocatalytic efficiency in order to have a proper design of the photocatalytic technology.' Please do this work for us in the review paper. I would like to be able to look up the energy consumption in your table. The tables list 'efficiency', which means removal efficiency of the system, but I think this has very little meaning without some normalizing parameter such as power input or surface area or time. In fact, the numbers are impossible to compare without consideration of these other factors. If you say this balance must be considered than please do so - give the example.

Overall -- a good start -- and quality should be improved before publication, please revise.

Reviewer 2 Report

The review prepared by Enesca et.al reviewed the heterostructure photocatalysts for dye and pharmaceutical active compounds degradation. The general comment is that the author did not provide their insights on this topic but only listing some literatures. For example, the authors list the 4 types of heterostructure photocatalysts in Fig.2 and 3, but did not discuss their roles for dye or pharmaceutical compound degradation process in Sections 3. Besides, some critical points on this topic are not discussed. Recently there are many examples by combining adsorption and photocatalysis for dye and pharmaceutical active compounds degradation. This emerging field was not discussed. Moreover, the new type of heterostructure (S-scheme) was not mentioned in this review. For the figures, the authors need to provide images of some photocatalysts, which is helpful for the readers to understand. For Figure 1, it seems the authors want to highlight the solar radiation spectrum as a background. But it is hard to read it at the current form. With the above issues, I would not recommend it for publication in Nanomaterials.

Reviewer 3 Report

The manuscript is a mini-review on the use of heterostructures as photocatalysts for the abatement of dyes and pharmaceutical pollutants. The content of the manuscript is largely insufficient as a review, since it presents a list of materials and applications without attempting a rational comment of the results or systematic investigation on the different applications.

From the very beginning, for instance, target pollutants are selected without looking at the stringent environmental problem (e.g. which are the most used/spread in the environment) pollutants. The selection starts with model molecules reported in the literature, such as MG or MO, that are often selected as models only because they are widely available, manageable easily or easy to analyse. So on this point the practical interest of the review is somehow limited.

Furthermore, the selected examples are reported as a summary of performance, a list of applications has been provided but without any attempt of rationalisation or in depth interpretation of their action. Also the discussion on the different types of heterostructure is lost, merging together the different examples and avoiding any sound conclusion on the best approach.

In the conclusion the authors cite a balance between energy consumption and activity, but a sound discussion on the energy consumption is actually missing.

Many recent reports in the field are actually missing, finding instead a lot of citations by the authors.

Finally, many typos are resent, such as:

Line 53: one more

Line 209: photocatalytice efficiency

Based on this, I cannot recommend this mini-review for publication.